# Exact sampling of determinantal point processes with sublinear time preprocessing

**Michał Dereziński**[*]
Department of Statistics
University of California, Berkeley
mderezin@berkeley.edu

**Daniele Calandriello**[*]
LCSL
Istituto Italiano di Tecnologia, Italy
daniele.calandriello@iit.it

**Michal Valko**
DeepMind Paris
valkom@deepmind.com

## Abstract

We study the complexity of sampling from a distribution over all index subsets of the set $\{1, ..., n\}$ with the probability of a subset $S$ proportional to the determinant of the submatrix $\mathbf{L}_S$ of some $n \times n$ positive semidefinite matrix $\mathbf{L}$, where $\mathbf{L}_S$ corresponds to the entries of $\mathbf{L}$ indexed by $S$. Known as a determinantal point process (DPP), this distribution is used in machine learning to induce diversity in subset selection. When sampling from DDPs, we often wish to sample *multiple* subsets $S$ with small expected size $k \triangleq \mathbb{E}[|S|] \ll n$ from a very large matrix $\mathbf{L}$, so it is important to minimize the preprocessing cost of the procedure (performed once) as well as the sampling cost (performed repeatedly). For this purpose we provide DPP-VFX, a new algorithm which, given access *only* to $\mathbf{L}$, samples *exactly* from a determinantal point process while satisfying the following two properties: (1) its preprocessing cost is $n \cdot \mathrm{poly}(k)$, i.e., *sublinear* in the size of $\mathbf{L}$, and (2) its sampling cost is $\mathrm{poly}(k)$, i.e., *independent* of the size of $\mathbf{L}$. Prior to our results, state-of-the-art exact samplers required $\mathcal{O}(n^3)$ preprocessing time and sampling time linear in $n$ or dependent on the spectral properties of $\mathbf{L}$. We furthermore give a reduction which allows using our algorithm for *exact* sampling from cardinality constrained determinantal point processes with $n \cdot \mathrm{poly}(k)$ time preprocessing. Our implementation of DPP-VFX is provided at https://github.com/guilgautier/DPPy/.

## 1 Introduction

Given a positive semi-definite (psd) $n \times n$ matrix $\mathbf{L}$, a determinantal point process $\mathrm{DPP}(\mathbf{L})$, also known as an $L$-ensemble, is a distribution over all $2^n$ index subsets $S \subseteq \{1, \ldots, n\}$ such that

$$\Pr(S) \triangleq \frac{\det(\mathbf{L}_S)}{\det(\mathbf{I} + \mathbf{L})},$$

where $\mathbf{L}_S$ denotes the $|S| \times |S|$ submatrix of $\mathbf{L}$ with rows and columns indexed by $S$. Determinantal point processes naturally appear across many scientific domains [Mac75, BLMV17, Gue83], and they have emerged as an important tool in machine learning [KT12] for inducing diversity in subset selection and as a variance reduction approach. DPP sampling has been successfully applied in core ML problems such as recommender systems [GKVM18, CZZ18, Bru18], stochastic optimization [ZKM17, ZÖMS19], data summarization [CKS+18], Gaussian processes [MS16, BRVDW19],

---

[*]Equal contribution.

| | exact | DPP | $k$-DPP | first sample | subsequent samples |
|---|---|---|---|---|---|
| [HKP+06, KT11] | ✓ | ✓ | ✓ | $n^3$ | $nk^2$ |
| [AGR16] | ✗ | ✗ | ✓ | $n \cdot \mathrm{poly}(k)$ | $n \cdot \mathrm{poly}(k)$ |
| [LJS16b] | ✗ | ✓ | ✗ | $n^2 \cdot \mathrm{poly}(k)$ | $n^2 \cdot \mathrm{poly}(k)$ |
| [LGD18] | ✓ | ✓ | ✗ | $n^3$ | $\mathrm{poly}(k \cdot (1 + \|\mathbf{L}\|))$ |
| [Der19] | ✓ | ✓ | ✗ | $n^3$ | $\mathrm{poly}(\mathrm{rank}(\mathbf{L}))$ |
| DPP-VFX **(this paper)** | ✓ | ✓ | ✓ | $n \cdot \mathrm{poly}(k)$ | $\mathrm{poly}(k)$ |

Table 1: Comparison of DPP and k-DPP algorithms using the $L$-ensemble representation. For a DPP, $k$ denotes the expected subset size. Note that $k \leq \mathrm{rank}(\mathbf{L}) \leq n$. We omit log terms for clarity.

experimental design [DW17, DWH18], and many more. In these applications, we often wish to efficiently produce many DPP samples of small expected size[2] $k \triangleq \mathbb{E}[|S|]$ given a large matrix $\mathbf{L}$. Sometimes, the distribution is restricted to subsets of fixed size $|S| = k \ll n$, denoted $k$-DPP($\mathbf{L}$). [HKP+06] gave an algorithm for drawing samples from DPP($\mathbf{L}$) distributed exactly, later adapted to $k$-DPP($\mathbf{L}$) by [KT11], which can be implemented to run in polynomial time. In many applications, however, sampling is still a computational bottleneck because the algorithm requires performing the eigendecomposition of matrix $\mathbf{L}$ at the cost of $\mathcal{O}(n^3)$. In addition to that initial cost, producing many independent samples $S_1, S_2, \ldots$ at high frequency poses a challenge because the cost of each sample is at least linear in $n$. Many alternative algorithms exist for both DPPs and $k$-DPPs to reduce the computational cost of preprocessing and/or sampling, including many approximate and heuristic approaches. Contrary to approximate solutions, we present an algorithm which samples *exactly* from a DPP or a $k$-DPP with the initial preprocessing cost *sublinear* in the size of $\mathbf{L}$ and the sampling cost *independent* of the size of $\mathbf{L}$.

**Theorem 1** *For a psd $n \times n$ matrix $\mathbf{L}$, let $S_1, S_2, \ldots$ be i.i.d. random sets from DPP($\mathbf{L}$) or from any $k$-DPP($\mathbf{L}$). Then, there is an algorithm which, given access to $\mathbf{L}$, returns*

    *a) the first subset, $S_1$, in:*    $n \cdot \mathrm{poly}(k)\, \mathrm{polylog}(n)$ *time,*

    *b) each subsequent $S_i$ in:*      $\mathrm{poly}(k)$ *time.*

We refer to this algorithm as the Very Fast and eXact DPP sampler, or DPP-VFX. Table 1 compares DPP-VFX with other DPP and $k$-DPP sampling algorithms. In this comparison, we feature the methods that provide strong accuracy guarantees. As seen from the table, our algorithm is the first exact sampler to achieve sublinear overall runtime. Only the approximate MCMC sampler of [AGR16] matches our $n \cdot \mathrm{poly}(k)$ complexity (and only for a $k$-DPP), but for this method every next sample is equally expensive, making it less practical when repeated sampling is needed. In fact, to our knowledge, no other exact or approximate method (with rigorous approximation guarantees) achieves $\mathrm{poly}(k)$ sampling time of the present paper.

Our method is based on a technique developed recently by [DWH18, DWH19] and later extended by [Der19]. In this approach, we carefully downsample the index set $[n] = \{1, ..., n\}$ to a sample $\sigma = (\sigma_1, ..., \sigma_t) \in [n]^t$ that is small but still sufficiently larger than the expected target size $k$, and then run a DPP on $\sigma$. As the downsampling distribution we use a *regularized* determinantal point process (R-DPP), proposed by [Der19], which (informally) samples $\sigma$ with probability $\Pr(\sigma) \sim \det(\mathbf{I} + \widetilde{\mathbf{L}}_\sigma)$, where $\widetilde{\mathbf{L}}$ is a rescaled version of $\mathbf{L}$. We can summarize this approach as follows, where $|S| \leq t \ll n$,

$$\{1, ..., n\} \xrightarrow{\sigma \sim \text{R-DPP}} (\sigma_1, ..., \sigma_t) \xrightarrow{\widetilde{S} \sim \text{DPP}} S = \{\sigma_i : i \in \widetilde{S}\}.$$

The DPP algorithm proposed by [Der19] follows the same diagram, however it requires that the size of the intermediate sample $\sigma$ be $\Omega(\mathrm{rank}(\mathbf{L}) \cdot k)$. This means that their method provides improvement over [HKP+06] only when $\mathbf{L}$ can be decomposed as $\mathbf{X}\mathbf{X}^\top$ for some $n \times r$ matrix $\mathbf{X}$, with $r \ll n$. However, in practice, matrix $\mathbf{L}$ is often only *approximately low-rank*, i.e., it exhibits some form of eigenvalue decay but it does not have a low-rank factorization. In this case, the results of [Der19] are vacuous both in terms of the preprocessing cost and the sampling cost, in that obtaining *every* sample would take $\Omega(n^3)$. We propose a different R-DPP implementation (see DPP-VFX as Algorithm 1)

where the expected size of $\sigma$ is $\mathcal{O}(k^2)$. To make the algorithm efficient, we use new connections between determinantal point processes, ridge leverage scores, and Nyström approximation.

**Definition 1** *Given a psd matrix $\mathbf{L}$, its ith $\lambda$-ridge leverage score (RLS) $\tau_i(\lambda)$ is the ith diagonal entry of $\mathbf{L}(\lambda \mathbf{I} + \mathbf{L})^{-1}$. The $\lambda$-effective dimension $d_{\mathrm{eff}}(\lambda)$ is the sum of the leverage scores, $\sum_i \tau_i(\lambda)$.*

An important connection between RLSs and DPPs is that when $S \sim \mathrm{DPP}(\mathbf{L})$, the marginal probability of index $i$ being sampled into $S$ is equal to the $i$th 1-ridge leverage score of $\mathbf{L}$, and the expected size $k$ of $S$ is equal to the 1-effective dimension,

$$\Pr(i \in S) = \big[\mathbf{L}(\mathbf{I} + \mathbf{L})^{-1}\big]_{ii} = \tau_i(1), \qquad k \triangleq \mathbb{E}\big[|S|\big] = \mathrm{tr}\big(\mathbf{L}(\mathbf{I} + \mathbf{L})^{-1}\big) = d_{\mathrm{eff}}(1).$$

Intuitively, if the marginal probability of $i$ is high, then this index should likely make it into the intermediate sample $\sigma$. This suggests that i.i.d. sampling of the indices $\sigma_1, ..., \sigma_t$ proportionally to 1-ridge leverage scores, i.e., $\Pr(\sigma_1 = i) \propto \tau_i(1)$, should serve as a reasonable and cheap heuristic for constructing $\sigma$. In fact, we can show that this distribution can be easily corrected by rejection sampling to become the R-DPP that we need. Computing ridge leverage scores exactly costs $\mathcal{O}(n^3)$, so instead we compute them approximately by first constructing a Nyström approximation of $\mathbf{L}$.

**Definition 2** *Let $\mathbf{L}$ be a psd matrix and $C$ a subset of its row/column indices with size $m \triangleq |C|$. Then we define the Nyström approximation of $\mathbf{L}$ based on $C$ as the $n \times n$ matrix $\widehat{\mathbf{L}} \triangleq (\mathbf{L}_{C,\mathcal{I}})^\top \mathbf{L}_C^+ \mathbf{L}_{C,\mathcal{I}}$.*

Here, $\mathbf{L}_{C,\mathcal{I}}$ denotes an $m \times n$ matrix consisting of (entire) rows of $\mathbf{L}$ indexed by $C$ and $(\cdot)^+$ denotes the Moore-Penrose pseudoinverse. Since we use rejection sampling to achieve the right intermediate distribution, the correctness of our algorithm does not depend on which Nyström approximation is chosen. However, the subset $C$ greatly influences the computational cost of the sampling through the rank of $\widehat{\mathbf{L}}$ and the probability of rejecting a sample. Since $\mathrm{rank}(\widehat{\mathbf{L}}) = |C|$, operations such as multiplication and inversion involving the Nyström approximation will scale with $m$, and therefore a small subset increases efficiency. However, if $\widehat{\mathbf{L}}$ is too different from $\mathbf{L}$, the probability of rejecting the sample will be very high and the algorithm inefficient. In this case, a slightly larger subset could improve accuracy and acceptance rate without increasing too much the cost of handling $\widehat{\mathbf{L}}$. Therefore, subset $C$ has to be selected so that it is both small and accurately represents the matrix $\mathbf{L}$. Here, we once again rely on ridge leverage score sampling which has been effectively used for obtaining good Nyström approximations in a number of prior works such as [AM15, CLV17, RCCR18].

While our main algorithm can sample only from the *random-size* DPP, and not from the *fixed-size* $k$-DPP, we present a rigorous reduction argument which lets us use our DPP algorithm to sample exactly from a $k$-DPP (for any $k$) with a small computational overhead.

**Related work** Prior to our work, fast exact sampling from generic DPPs has been considered out of reach. The first procedure to sample general DPPs was given by [HKP$^+$06] and even most recent exact refinements [LGD18, Der19, Pou19], when the DPP is represented in the form of an $L$-ensemble, require preprocessing that amounts to an expensive $n \times n$ matrix diagonalization at the cost $\mathcal{O}(n^3)$, which is shown as the *first-sample* complexity column in Table 1.

Nonetheless, there are well-known samplers for very specific DPPs that are both fast and exact, for instance for sampling uniform spanning trees [Ald90, Bro89, PW98], which leaves the possibility of a more generic fast sampler open. Since the sampling from DPPs has several practical large scale machine learning applications [KT12], there are now a number of methods known to be able to sample from a DPP *approximately*, outlined in the following paragraphs.

As DPPs can be specified by kernels ($L$-kernels or $K$-kernels), a natural approximation strategy is to resort to low-rank approximations [KT11, GKT12, AKFT13, LJS16a]. For example, [AKFT13] provides approximate guarantee for the probability of any subset being sampled as a function of eigengaps of the $L$-kernel. Next, [LJS16a] construct *coresets* approximating a given $k$-DPP and then use them for sampling. In their Section 4.1, [LJS16a] show in which cases we can hope for a good approximation. These guarantees become tight if these approximations (Nyström subspace, coresets) are aligned with data. In our work, we aim for an *adaptive* approach that is able to provide a good approximation for *any* DPP.

The second class of approaches are based on Markov chain Monte-Carlo [MU49] techniques [Kan13, RK15, AGR16, LJS16b, GBV17]. There are known polynomial bounds on the mixing rates [DS91]

of MCMC chains with arbitrary DPPs as their limiting measure. In particular, [AGR16] showed them for cardinality-constrained DPPs and [LJS16b] for the general case. The two chains have mixing times which are, respectively, linear and quadratic in $n$ (see Table 1). Unfortunately, for any subsequent sample we need to wait until the chain *mixes again*.

Neither the known low-rank approximations or the known MCMC methods are able to provide samples that are *exactly* distributed according to a DPP (also called *perfect sampling*). This is not surprising as having *scalable and exact* sampling is very challenging in general. For example, methods based on rejection sampling are *always exact*, but they typically scale poorly to high-dimensional data and are adversely affected by the spikes in the distribution [EVCM16], resulting in high rejection rate and inefficiency. Surprisingly, our method is based on both low-rank approximation (a source of inaccuracy) and rejection sampling (a common source of inefficiency). In the following section, we show how to obtain a *perfect DPP sampler* from a Nyström approximation of the $L$-kernel. Then, to guarantee efficiency, in Section 3 we bound the number of rejections, which is possible thanks to the use of *intermediate downsampling*.

## 2 Exact sampling using any Nyström approximation

**Notation**  We use $[n]$ to denote the set $\{1, \ldots, n\}$. For a matrix $\mathbf{B} \in \mathbb{R}^{n \times m}$ and index sets $C$, $D$, we use $\mathbf{B}_{C,D}$ to denote the submatrix of $\mathbf{B}$ consisting of the intersection of rows indexed by $C$ with columns indexed by $D$. If $C = D$, we use a shorthand $\mathbf{B}_C$ and if $D = [m]$, we may write $\mathbf{B}_{C,\mathcal{I}}$. Finally, we also allow $C, D$ to be multisets or sequences, in which case each row/column is duplicated in the matrix according to its multiplicity (and in the case of sequences, we order the rows/columns as they appear in the sequence). Note that with this notation if $\mathbf{L} = \mathbf{B}\mathbf{B}^\top$ then $\mathbf{L}_{C,D} = \mathbf{B}_{C,\mathcal{I}}\mathbf{B}_{D,\mathcal{I}}^\top$.

As discussed in the introduction, our method relies on an intermediate downsampling distribution to reduce the size of the problem. The exactness of our sampler relies on the careful choice of that intermediate distribution. To that end, we use *regularized* determinantal processes, introduced by [Der19]. In the below definition, we adapt them to the kernel setting.

**Definition 3**  *Given an $n \times n$ psd matrix $\mathbf{L}$, distribution $p \triangleq (p_1, \ldots, p_n)$ and $r > 0$, let $\widetilde{\mathbf{L}}$ denote an $n \times n$ matrix such that $\widetilde{L}_{ij} \triangleq \frac{1}{r\sqrt{p_i p_j}} L_{ij}$ for all $i, j \in [n]$. We define $\mathrm{R\text{-}DPP}_p^r(\mathbf{L})$ as a distribution over events $A \subseteq \bigcup_{k=0}^{\infty} [n]^k$, where*

$$\Pr(A) \triangleq \frac{\mathbb{E}_\sigma\left[\mathbf{1}_{[\sigma \in A]} \det(\mathbf{I} + \widetilde{\mathbf{L}}_\sigma)\right]}{\det(\mathbf{I} + \mathbf{L})}, \quad \textit{for} \quad \sigma = (\sigma_1, \ldots, \sigma_t) \overset{\text{i.i.d.}}{\sim} p, \quad t \sim \mathrm{Poisson}(r).$$

Since the term $\det(\mathbf{I} + \widetilde{\mathbf{L}}_\sigma)$ has the same form as the normalization constant of $\mathrm{DPP}(\widetilde{\mathbf{L}}_\sigma)$, an easy calculation shows that the R-DPP can be used as an intermediate distribution in our algorithm without introducing any distortion in the sampling.

**Proposition 1 (Der19, Theorem 8)**  *For any $\mathbf{L}$, $p$, $r$, and $\widetilde{\mathbf{L}}$ defined as in Definition 3,*

$$\textit{if} \quad \sigma \sim \mathrm{R\text{-}DPP}_p^r(\mathbf{L}) \quad \textit{and} \quad S \sim \mathrm{DPP}(\widetilde{\mathbf{L}}_\sigma) \quad \textit{then} \quad \{\sigma_i : i \in S\} \sim \mathrm{DPP}(\mathbf{L}).$$

To sample from the R-DPP, our algorithm uses rejection sampling, where the proposal distribution is sampling i.i.d. proportionally to the approximate 1-ridge leverage scores $l_i \approx \tau_i(1)$ (see Definition 1 and the following discussion), computed using any Nyström approximation $\widehat{\mathbf{L}}$ of matrix $\mathbf{L}$. Apart from $\widehat{\mathbf{L}}$, the algorithm also requires an additional parameter $q$, which controls the size of the intermediate sample. Because of rejection sampling and Proposition 1, the correctness of the algorithm *does not depend* on the choice of $\widehat{\mathbf{L}}$ and $q$, as demonstrated in the following result. The key part of the proof involves showing that the acceptance probability in Line 6 is bounded by 1. Here, we obtain a considerably tighter bound than the one achieved by [Der19], which allows us to use a much smaller intermediate sample $\sigma$ (see Section 3) while maintaining the efficiency of rejection sampling.

**Theorem 2**  *Given a psd matrix $\mathbf{L}$, any one of its Nyström approximations $\widehat{\mathbf{L}}$ and any positive $q$, DPP-VFX (Algorithm 1) returns $S \sim \mathrm{DPP}(\mathbf{L})$.*

An important implication of Theorem 2 is that even though the choice of $\widehat{\mathbf{L}}$ affects the overall execution of the algorithm, it *does not affect the distribution of the output*. Therefore we can reuse

---

**Algorithm 1** DPP-VFX sampling $S \sim \mathrm{DPP}(\mathbf{L})$

---

**Input:** $\mathbf{L} \in \mathbb{R}^{n \times n}$, its Nyström approximation $\widehat{\mathbf{L}}$, $q > 0$

1: Compute $l_i = \big[ (\mathbf{L} - \widehat{\mathbf{L}}) + \widehat{\mathbf{L}}(\mathbf{I} + \widehat{\mathbf{L}})^{-1} \big]_{ii} \approx \Pr(i \in S)$

2: Initialize $s = \sum_i l_i, \quad z = \mathrm{tr}\big( \widehat{\mathbf{L}}(\mathbf{I} + \widehat{\mathbf{L}})^{-1} \big), \quad \widetilde{\mathbf{L}} = \frac{s}{q} \Big[ \frac{1}{\sqrt{l_i l_j}} L_{ij} \Big]_{ij}$

3: **repeat**

4:   sample $t \sim \mathrm{Poisson}(q\,\mathrm{e}^{s/q})$

5:   sample $\sigma_1, \dots, \sigma_t \overset{\mathrm{i.i.d.}}{\sim} \big( \frac{l_1}{s}, \dots, \frac{l_n}{s} \big)$,

6:   sample $Acc \sim \mathrm{Bernoulli}\Big( \frac{\mathrm{e}^z \det(\mathbf{I} + \widetilde{\mathbf{L}}_\sigma)}{\mathrm{e}^{ts/q} \det(\mathbf{I} + \widehat{\mathbf{L}})} \Big)$

7: **until** $Acc = \mathrm{true}$

8: sample $\widetilde{S} \sim \mathrm{DPP}(\widetilde{\mathbf{L}}_\sigma)$

9: **return** $S = \{\sigma_i : i \in \widetilde{S}\}$

---

the same $\widehat{\mathbf{L}}$ to produce multiple independent samples $S_1, S_2, \dots \sim \mathrm{DPP}(\mathbf{L})$, which we prove in Appendix A. This is particularly important because, as we will see in Section 3, DPP-VFX can generate successive samples $S_2, S_3, \dots$ much more cheaply than the first sample $S_1$.

**Lemma 1** *Let $C \subseteq [n]$ be a random set variable with any distribution. Suppose that $S_1$ and $S_2$ are returned by two executions of* DPP-VFX*, both using inputs constructed from the same $\mathbf{L}$ and $\widehat{\mathbf{L}} = \mathbf{L}_{\mathcal{I},C} \mathbf{L}_C^+ \mathbf{L}_{C,\mathcal{I}}$. Then $S_1$ and $S_2$ are (unconditionally) independent.*

Before proceeding with the proof, we highlight for clarity that the Poisson r.v. $t$ in DPP-VFX (Algorithm 1, Line 4) has a different role than the the Poisson r.v. used in the definition of R-DPPs. As a result, the definition of $\widetilde{\mathbf{L}}$ (Line 2) is not exactly the one that would be suggested by Definition 3. In particular, DPP-VFX uses $q\,\mathrm{e}^{s/q}$ instead of just $q$ as the mean parameter for $t$. The extra $\mathrm{e}^{s/q}$ factor is *necessary* to correct for the rejection sampling step in Line 6 during the sampling loop, see the proof below.

**Proof of Theorem 2**   We start by showing that the Bernoulli probability in Line 6 is bounded by 1. Note that this is important not only to sample correctly, but also when we later establish the efficiency of the algorithm. If we showed a weaker upper bound, say $c > 1$, we could always divide the expression by $c$ and retain the correctness, however it would also be $c$ times less likely that $Acc = 1$.

Since $\widehat{\mathbf{L}}$ is a Nyström approximation for some $C \subseteq \mathcal{I}$, it can be written as

$$\widehat{\mathbf{L}} = \mathbf{L}_{\mathcal{I},C} \mathbf{L}_C^+ \mathbf{L}_{C,\mathcal{I}} = \mathbf{B}\mathbf{B}_{C,\mathcal{I}}^\top \mathbf{L}_C^+ \mathbf{B}_{C,\mathcal{I}} \mathbf{B}^\top = \mathbf{B}\mathbf{P}\mathbf{B}^\top,$$

for any $\mathbf{B}$ such that $\mathbf{L} = \mathbf{B}\mathbf{B}^\top$ where $\mathbf{P} \triangleq \mathbf{B}_{C,\mathcal{I}}^\top \mathbf{L}_C^+ \mathbf{B}_{C,\mathcal{I}}$ is a projection, so that $\mathbf{P}^2 = \mathbf{P}$. Let $\widetilde{\mathbf{L}} \triangleq \widetilde{\mathbf{B}}\widetilde{\mathbf{B}}^\top$ where the $i$th row of $\widetilde{\mathbf{B}}$ is the rescaled $i$th row of $\mathbf{B}$, i.e. $\widetilde{\mathbf{b}}_i^\top \triangleq \sqrt{\frac{s}{q l_i}}\,\mathbf{b}_i^\top$. Then, we have

$$\frac{\det(\mathbf{I} + \widetilde{\mathbf{L}}_\sigma)}{\det(\mathbf{I} + \widehat{\mathbf{L}})} = \frac{\det(\mathbf{I} + \widetilde{\mathbf{B}}_{\sigma,\mathcal{I}}^\top \widetilde{\mathbf{B}}_{\sigma,\mathcal{I}})}{\det(\mathbf{I} + \mathbf{P}\mathbf{B}^\top \mathbf{B}\mathbf{P})} = \det\big( (\mathbf{I} + \widetilde{\mathbf{B}}_{\sigma,\mathcal{I}}^\top \widetilde{\mathbf{B}}_{\sigma,\mathcal{I}})(\mathbf{I} + \mathbf{P}\mathbf{B}^\top \mathbf{B}\mathbf{P})^{-1} \big)$$

$$= \det\big( \mathbf{I} + (\widetilde{\mathbf{B}}_{\sigma,\mathcal{I}}^\top \widetilde{\mathbf{B}}_{\sigma,\mathcal{I}} - \mathbf{P}\mathbf{B}^\top \mathbf{B}\mathbf{P})(\mathbf{I} + \mathbf{P}\mathbf{B}^\top \mathbf{B}\mathbf{P})^{-1} \big)$$

$$\leq \exp\Big( \mathrm{tr}\big( (\widetilde{\mathbf{B}}_{\sigma,\mathcal{I}}^\top \widetilde{\mathbf{B}}_{\sigma,\mathcal{I}} - \mathbf{P}\mathbf{B}^\top \mathbf{B}\mathbf{P})(\mathbf{I} + \mathbf{P}\mathbf{B}^\top \mathbf{B}\mathbf{P})^{-1} \big) \Big)$$

$$= \exp\Big( \sum_{i=1}^t \frac{s}{q l_{\sigma_i}} \big[ \mathbf{B}(\mathbf{I} + \mathbf{P}\mathbf{B}^\top \mathbf{B}\mathbf{P})^{-1} \mathbf{B}^\top \big]_{\sigma_i \sigma_i} \Big) \cdot \mathrm{e}^{-z} = \mathrm{e}^{ts/q} \mathrm{e}^{-z},$$

where the last equality follows because

$$\mathbf{B}(\mathbf{I} + \mathbf{P}\mathbf{B}^\top \mathbf{B}\mathbf{P})^{-1} \mathbf{B}^\top = \mathbf{B}\big( \mathbf{I} - \mathbf{P}\mathbf{B}^\top (\mathbf{I} + \mathbf{B}\mathbf{P}\mathbf{B}^\top)^{-1} \mathbf{B}\mathbf{P} \big) \mathbf{B}^\top$$

$$= \mathbf{L} - \widehat{\mathbf{L}}(\mathbf{I} + \widehat{\mathbf{L}})^{-1}\widehat{\mathbf{L}} = \mathbf{L} - \widehat{\mathbf{L}} + \widehat{\mathbf{L}}(\mathbf{I} + \widehat{\mathbf{L}})^{-1}.$$

Thus, we showed that the expression in Line 6 is valid. Let $\widetilde{\sigma}$ denote the random variable distributed as $\sigma$ is after exiting the repeat loop. It follows that

$$\Pr(\widetilde{\sigma} \in A) \propto \mathbb{E}_\sigma\left[\mathbf{1}_{[\sigma \in A]} \frac{\mathrm{e}^z \det(\mathbf{I} + \widetilde{\mathbf{L}}_\sigma)}{\mathrm{e}^{ts/q} \det(\mathbf{I} + \widehat{\mathbf{L}})}\right] \propto \sum_{t=0}^{\infty} \frac{(q\,\mathrm{e}^{s/q})^t}{\mathrm{e}^{q\,\mathrm{e}^{s/q}} t!} \cdot \mathrm{e}^{-ts/q}\, \mathbb{E}_\sigma\left[\mathbf{1}_{[\sigma \in A]} \det(\mathbf{I} + \widetilde{\mathbf{L}}_\sigma) \mid t\right]$$

$$\propto \mathbb{E}_{t'}\left[\mathbb{E}_\sigma\left[\mathbf{1}_{[\sigma \in A]} \det(\mathbf{I} + \widetilde{\mathbf{L}}_\sigma) \mid t = t'\right]\right] \quad \text{for } t' \sim \mathrm{Poisson}(q),$$

which shows that $\widetilde{\sigma} \sim \text{R-DPP}_l^q(\mathbf{L})$ for $l = (\frac{l_1}{s}, \dots, \frac{l_n}{s})$. The claim follows from Proposition 1. ∎

## 3   Conditions for fast sampling

The complexity cost of DPP-VFX can be roughly summarized as follows: we pay a large one-time cost to precompute $\widehat{\mathbf{L}}$ and all its associated quantities, and then we pay a smaller cost in the rejection sampling scheme which must be multiplied by the number of times we repeat the loop until acceptance. We first show that if $\widehat{\mathbf{L}}$ is sufficiently close to $\mathbf{L}$ then we will exit the loop with high probability. We then analyze how accurate the precomputing step needs to be to satisfy this condition. Finally, we bound the overall computational cost, which includes also the final step where we sample $S$ out of the intermediate sample $\sigma$ using any off-the-shelf exact DPP sampler.

**1) Bounding the number of rejections** The following result presents the two conditions needed for achieving efficient rejection sampling in DPP-VFX. First, the Nyström approximation needs to be accurate enough, and second, the intermediate sample size (controlled by parameter $q$) needs to be $\Omega(k^2)$. This is a significant improvement over the guarantee of [Der19] where the intermediate sample size is $\Omega(\mathrm{rank}(\mathbf{L}) \cdot k)$, which is only meaningful for low-rank kernels. The main novelty in this proof comes from showing the following lower bound on the ratio of the determinants of $\mathbf{I} + \mathbf{L}$ and $\mathbf{I} + \widehat{\mathbf{L}}$, when $\widehat{\mathbf{L}}$ is a Nyström approximation: $\det(\mathbf{I} + \mathbf{L})/\det(\mathbf{I} + \widehat{\mathbf{L}}) \geq \mathrm{e}^{k-z}$, where $z \triangleq \mathrm{tr}(\widehat{\mathbf{L}}(\mathbf{I} + \widehat{\mathbf{L}})^{-1})$. Remarkably, this bound exploits the fact that any Nyström approximation of $\mathbf{L} = \mathbf{B}\mathbf{B}^\top$ can be written as $\widehat{\mathbf{L}} = \mathbf{B}\mathbf{P}\mathbf{B}^\top$, where $\mathbf{P}$ is a projection matrix. Note that while our result holds in the worst case, in deployement, the conditions on $\widehat{\mathbf{L}}$ and on $q$ can be considerably relaxed.

**Theorem 3** *If the Nyström approximation $\widehat{\mathbf{L}}$ and the intermediate sample size parameter $q$ satisfy*

$$\mathrm{tr}\big(\mathbf{L}(\mathbf{I} + \mathbf{L})^{-1}\mathbf{L} - \widehat{\mathbf{L}}(\mathbf{I} + \mathbf{L})^{-1}\widehat{\mathbf{L}}\big) \leq 1 \qquad \text{and} \qquad q \geq \max\{s^2, s\},$$

*then* $\Pr(Acc = true) \geq \mathrm{e}^{-2}$. *Therefore, with probability* $1 - \delta$, *Algorithm 1 exits the rejection sampling loop after at most* $\mathcal{O}(\log \delta^{-1})$ *iterations and, after precomputing all of the inputs, the time complexity of the rejection sampling loop is* $\mathcal{O}\big(k^6 \log \delta^{-1} + \log^4 \delta^{-1}\big)$.

**Proof** We first bound the number of rejections, and then discuss the complexity of each loop iteration. Let $\sigma$ be distributed as in Line 5. The probability of exiting the repeat loop at each iteration is

$$P \triangleq \mathbb{E}_\sigma\left[\frac{\mathrm{e}^z \det(\mathbf{I} + \widetilde{\mathbf{L}}_\sigma)}{\mathrm{e}^{ts/q} \det(\mathbf{I} + \widehat{\mathbf{L}})}\right] = \sum_{t=0}^{\infty} \frac{(q\,\mathrm{e}^{s/q})^t}{\mathrm{e}^{q\,\mathrm{e}^{s/q}} t!} \cdot \frac{\mathrm{e}^{z-ts/q}}{\det(\mathbf{I} + \widehat{\mathbf{L}})} \mathbb{E}_\sigma\left[\det(\mathbf{I} + \widetilde{\mathbf{L}}_\sigma) \mid t\right]$$

$$= \frac{\mathrm{e}^{q-q\,\mathrm{e}^{s/q}+z}}{\det(\mathbf{I} + \widehat{\mathbf{L}})} \sum_{t=0}^{\infty} \frac{q^t}{\mathrm{e}^q t!} \mathbb{E}_\sigma\left[\det(\mathbf{I} + \widetilde{\mathbf{L}}_\sigma) \mid t\right] = \mathrm{e}^{q-q\,\mathrm{e}^{s/q}+z} \frac{\det(\mathbf{I} + \mathbf{L})}{\det(\mathbf{I} + \widehat{\mathbf{L}})},$$

where the last equality follows because the infinite series computes the normalization constant of $\text{R-DPP}_l^q(\mathbf{L})$ given in Definition 3. If $s \geq 1$ then $q = s^2$ and the inequality $\mathrm{e}^x \leq 1 + x + x^2$ for $x \in [0, 1]$ implies that $q - q\,\mathrm{e}^{s/q} + z = s^2(1 - \mathrm{e}^{1/s} + 1/s) + z - s \geq -1 + z - s$. On the other hand, if $q = s \in [0, 1]$, then $q - q\mathrm{e}^{s/q} + z = (1 - \mathrm{e})s + z \geq -1 + z - s$. We proceed to lower bound the determinantal ratio. Here, let $\mathbf{L} = \mathbf{B}\mathbf{B}^\top$ and $\widehat{\mathbf{L}} = \mathbf{B}\mathbf{P}\mathbf{B}^\top$, where $\mathbf{P}$ is a projection matrix. Then,

$$\frac{\det(\mathbf{I} + \mathbf{L})}{\det(\mathbf{I} + \widehat{\mathbf{L}})} = \frac{\det(\mathbf{I} + \mathbf{B}^\top\mathbf{B})}{\det(\mathbf{I} + \mathbf{P}\mathbf{B}^\top\mathbf{B}\mathbf{P})} = \det\big(\mathbf{I} - (\mathbf{B}^\top\mathbf{B} - \mathbf{P}\mathbf{B}^\top\mathbf{B}\mathbf{P})(\mathbf{I} + \mathbf{B}^\top\mathbf{B})^{-1}\big)^{-1}$$

$$\geq \exp\Big(\mathrm{tr}\big((\mathbf{B}^\top\mathbf{B} - \mathbf{P}\mathbf{B}^\top\mathbf{B}\mathbf{P})(\mathbf{I} + \mathbf{B}^\top\mathbf{B})^{-1}\big)\Big)$$

$$= \exp\Big(\mathrm{tr}\big(\mathbf{B}(\mathbf{I} + \mathbf{B}^\top\mathbf{B})^{-1}\mathbf{B}^\top\big) - \mathrm{tr}\big(\mathbf{B}\mathbf{P}(\mathbf{I} + \mathbf{B}^\top\mathbf{B})^{-1}\mathbf{P}\mathbf{B}^\top\big)\Big),$$

where we can reformulate $k = \text{tr}\big(\mathbf{B}(\mathbf{I} + \mathbf{B}^\top\mathbf{B})^{-1}\mathbf{B}^\top\big)$ and the other elements in terms of $\widehat{\mathbf{L}}$ and $\mathbf{L}$ as

$$\exp\Big(k - \text{tr}\big(\mathbf{B}\mathbf{P}(\mathbf{I} + \mathbf{B}^\top\mathbf{B} - \mathbf{B}^\top\mathbf{B})(\mathbf{I} + \mathbf{B}^\top\mathbf{B})^{-1}\mathbf{P}\mathbf{B}^\top\big)\Big)$$
$$= \exp\Big(k - \text{tr}(\widehat{\mathbf{L}}) + \text{tr}\big(\mathbf{B}\mathbf{P}\mathbf{B}^\top\mathbf{B}(\mathbf{I} + \mathbf{B}^\top\mathbf{B})^{-1}\mathbf{P}\mathbf{B}^\top\big)\Big)$$
$$= \exp\Big(k - \text{tr}(\widehat{\mathbf{L}}) + \text{tr}\big(\mathbf{B}\mathbf{P}\mathbf{B}^\top(\mathbf{I} + \mathbf{B}\mathbf{B}^\top)^{-1}\mathbf{B}\mathbf{P}\mathbf{B}^\top\big)\Big)$$
$$= \exp\Big(k - \text{tr}(\widehat{\mathbf{L}}) + \text{tr}\big(\widehat{\mathbf{L}}(\mathbf{I} + \mathbf{L})^{-1}\widehat{\mathbf{L}}\big)\Big).$$

Putting all together,

$$\mathrm{e}^{q - q\,\mathrm{e}^{s/q} + z}\frac{\det(\mathbf{I} + \mathbf{L})}{\det(\mathbf{I} + \widehat{\mathbf{L}})} \geq \exp\Big(-1 + z - s + k - \text{tr}(\widehat{\mathbf{L}}) + \text{tr}\big(\widehat{\mathbf{L}}(\mathbf{I} + \mathbf{L})^{-1}\widehat{\mathbf{L}}\big)\Big),$$

and using the definitions $k = \text{tr}\big(\mathbf{L}(\mathbf{I} + \mathbf{L})^{-1}\big)$, $s = \text{tr}(\mathbf{L} - \widehat{\mathbf{L}} + \widehat{\mathbf{L}}(\widehat{\mathbf{L}} + \mathbf{I})^{-1})$, and $z = \text{tr}(\widehat{\mathbf{L}}(\widehat{\mathbf{L}} + \mathbf{I})^{-1})$ on the middle term $z - s + k - \text{tr}(\widehat{\mathbf{L}})$ we have

$$\text{tr}\big(\widehat{\mathbf{L}}(\widehat{\mathbf{L}} + \mathbf{I})^{-1} + \mathbf{L}(\mathbf{I} + \mathbf{L})^{-1} - \mathbf{L} + \widehat{\mathbf{L}} - \widehat{\mathbf{L}}(\widehat{\mathbf{L}} + \mathbf{I})^{-1} - \widehat{\mathbf{L}}\big)$$
$$= \text{tr}\big(\mathbf{L}(\mathbf{I} + \mathbf{L})^{-1} - \mathbf{L}\big) = \text{tr}\big(\mathbf{L}(\mathbf{I} + \mathbf{L})^{-1} - \mathbf{L}(\mathbf{I} + \mathbf{L})^{-1}(\mathbf{I} + \mathbf{L})\big) = -\text{tr}\big(\mathbf{L}(\mathbf{I} + \mathbf{L})^{-1}\mathbf{L}\big),$$

and therefore,

$$\mathrm{e}^{q - q\,\mathrm{e}^{s/q} + z}\frac{\det(\mathbf{I} + \mathbf{L})}{\det(\mathbf{I} + \widehat{\mathbf{L}})} \geq \exp\Big(-1 + \text{tr}\big(\widehat{\mathbf{L}}(\mathbf{I} + \mathbf{L})^{-1}\widehat{\mathbf{L}}\big) - \text{tr}\big(\mathbf{L}(\mathbf{I} + \mathbf{L})^{-1}\mathbf{L}\big)\Big),$$

and we obtain our condition. Thus, with probability $1 - \delta$, the main loop will be repeated $\mathcal{O}(\log \delta^{-1})$ times. We now quantify the cost of a single loop iteration. First note that since the number of samples $t_i$ drawn from $l$ in the $i$th iteration of the loop is a Poisson distributed random variable, a standard Poisson tail bound implies that with probability $1 - \delta$, all iterations will satisfy $t_i = \mathcal{O}(k^2 + \log \delta^{-1})$. Then, drawing the Poisson r.v. $t_i$ (Line 4) requires $\mathcal{O}(k^2 + \log \delta^{-1})$ time. Drawing $\sigma$ from the multinomial $\big(\frac{l_1}{s}, \dots, \frac{l_n}{s}\big)$ can be done by first sorting the $l_i$, which is a one-time $\mathcal{O}(n\log(n))$ cost, and then using specialized samplers which require $\mathcal{O}(1)$ time [BP12]. Therefore, the dominant cost is computing the determinant of the matrix $\mathbf{I} + \widetilde{\mathbf{L}}_\sigma$ in $\mathcal{O}(t_i^3)$, and the result follows. ∎

**2) Bounding the precompute cost** All that is left is to control the cost of the precomputation phase. We separate the analysis into two steps: how much it costs to choose $\widehat{\mathbf{L}}$ to satisfy the assumption of Theorem 3, and how much it costs to compute everything else given $\widehat{\mathbf{L}}$, see Appendix A.

**Lemma 2** *Let $\widehat{\mathbf{L}}$ be constructed by sampling $m = \mathcal{O}(k^3 \log \frac{n}{\delta})$ columns proportionally to their RLS. Then, with probability $1 - \delta$, $\widehat{\mathbf{L}}$ satisfies the assumption of Theorem 3.*

There exist many algorithms to sample columns proportionally to their RLSs. For example, we can take the BLESS [RCCR18] with the following guarantee.

**Proposition 2 (RCCR18, Theorem 1)** *There exists an algorithm that with probability $1 - \delta$ samples $m$ columns proportionally to their RLSs in $\mathcal{O}(nk^2 \log^2 \frac{n}{\delta} + k^3 \log^4 \frac{n}{\delta} + m)$ time.*

We can now compute the remaining preprocessing costs, given a Nyström approximation $\widehat{\mathbf{L}}$.

**Lemma 3** *Given $\widehat{\mathbf{L}}$ with rank $m$, we can compute $l_i$, $s$, $z$, and $\widetilde{\mathbf{L}}$ in $\mathcal{O}(nm^2 + m^3)$ time.*

**3) Bounding the overall cost** We can now fully characterize the computational cost.

**Theorem 1 (restated for DPPs only)** *For a psd $n \times n$ matrix $\mathbf{L}$, let $S_1, S_2$ be i.i.d. random sets from $\text{DPP}(\mathbf{L})$. Denote with $\widehat{\mathbf{L}}$ a Nyström approximation of $\mathbf{L}$ obtained by sampling $m = \mathcal{O}(k^3 \log(n/\delta))$ of its columns proportionally to their RLSs. If $q \geq \max\{s^2, s\}$, then w.p. $1 - \delta$, DPP-VFX returns*

a) *subset $S_1$ in:* $\mathcal{O}(nk^6 \log^2 \frac{n}{\delta} + k^9 \log^3 \frac{n}{\delta} + k^3 \log^4 \frac{n}{\delta})$ *time,*

b) *then, $S_2$ in:* $\mathcal{O}\big(k^6 \log \frac{1}{\delta} + \log^4 \frac{1}{\delta}\big)$ *time.*

**Discussion** The above result follows from the bounds on the precompute costs, on the number of iteratios, on the iterations cost, and on the fact that the final DPP sampling step $\widetilde{S} \sim \mathrm{DPP}(\widetilde{\mathbf{L}}_\sigma)$ requires $\mathcal{O}(t^3) \le \mathcal{O}((k^2 + \log \delta^{-1})^3)$ time. Note however that due to the nature of rejection sampling, as long as we exit the loop, i.e., we accept the sample, the output of DPP-VFX is guaranteed to follow the DPP distribution for any value of $m$ and $q$. In Theorem 1 we set $m = (\tilde{k}^3 \log \frac{n}{\delta})$ and $q \ge \max\{s^2, s\}$ to satisfy Theorem 3 and guarantee a constant acceptance probability in the rejection sampling loop, but this might not be necessary or even desirable in practice. Experimentally, much smaller values of $m$, starting from $m = \Omega(k \log \frac{n}{\delta})$ seem to be sufficient to accept the sample, while at the same time a smaller $m$ greatly reduces the preprocessing costs. In general, we recommend to separate DPP-VFX in three phases. First, compute an accurate estimate of the RLS using off-the-shelf algorithms in $\mathcal{O}(nk^2 \log^2 \frac{n}{\delta} + k^3 \log^4 \frac{n}{\delta})$ time. Then, sample a small number $m$ of columns to construct an explorative $\widehat{\mathbf{L}}$, and try to run DPP-VFX. If the rejection sampling loop does not terminate sufficiently fast, then we can reuse the RLS estimates to compute a more accurate $\widehat{\mathbf{L}}$ for a larger $m$. Using a simple doubling schedule for $m$, this strategy quickly reaches a regime where DPP-VFX is w.h.p. guaranteed to accept, resulting in faster sampling.

## 4   Reduction from DPPs to k-DPPs

We next show that with a simple extra rejection sampling step we can efficiently transform *any* exact DPP sampler into an exact $k$-DPP sampler.

A common heuristic to sample $S$ from a $k$-DPP is to first sample $S$ from a DPP, and then reject the sample if the size of $S$ is not exactly $k$. As we show in this section, the success probability of this procedure can be improved by appropriately rescaling $\mathbf{L}$ by a constant factor $\alpha$,

$$\text{sample} \quad S_\alpha \sim \mathrm{DPP}(\alpha \mathbf{L}), \quad \text{accept if } |S_\alpha| = k.$$

Note that rescaling the DPP by a constant $\alpha$ as above only changes the expected size of the set $S_\alpha$, and not its distribution. Therefore, if we accept only sets with size $k$, we will be sampling exactly from our $k$-DPP. Moreover, if $k_\alpha = \mathbb{E}[|S_\alpha|]$ is close to $k$, the success probability will improve. With a slight abuse of notation, in the context of $k$-DPPs we will indicate with $k$ the desired size of $S_\alpha$, i.e., the final output size at acceptance, and with $k_\alpha = \mathbb{E}[|S_\alpha|]$ the expected size of the scaled DPP.

While the above rejection sampling heuristic is widespread, until now there has been no proof that this heuristic can provably succeed in few rejections We solve this open question with two new results. First, we show that for an appropriate rescaling $\alpha^\star$ we only reject $S_{\alpha^\star}$ roughly $O(\sqrt{k})$ times. Then, we show how to find such an $\alpha^\star$ with a $\widetilde{\mathcal{O}}(n \cdot \mathrm{poly}(k))$ time preprocessing step.

**Theorem 4** *There exists constant $C > 0$ such that for any rank $n$ psd matrix $\mathbf{L}$ and $k \in [n]$, there is $\alpha^\star > 0$ with the following property: if we sample $S_{\alpha^\star} \sim \mathrm{DPP}(\alpha^\star \mathbf{L})$, then $\Pr(|S_{\alpha^\star}| = k) \ge \frac{1}{C\sqrt{k}}$.*

The proof in Appendix B. relies on a known Chernoff bound for the sample size $|S_\alpha|$ of a DPP. When applied naïvely, the inequality does not offer a lower bound on the probability of any single sample size. However we show that the probability mass is concentrated on $O(\sqrt{k_\alpha})$ sizes. This leads to a lower bound on the sample size with the largest probability, i.e., the *mode* of the distribution. Then, it remains to observe that for any $k \in [n]$ we can always find $\alpha^\star$ for which $k$ is the mode of $|S_{\alpha^\star}|$. We conclude that given $\alpha^\star$, the rejection sampling scheme described above transforms any $\mathrm{poly}(k)$ time DPP sampler into a $\mathrm{poly}(k)$ time $k$-DPP sampler. It remains to efficiently find $\alpha^\star$, which once again relies on using a Nyström approximation of $\mathbf{L}$.

**Lemma 4** *If $k \ge 1$, then there is an algorithm that finds $\alpha^\star$ in $\widetilde{\mathcal{O}}(n \cdot \mathrm{poly}(k))$ time.*

While the existence proof of $\alpha^\star$ is general and based on simple unimodality, this characterization is not sufficient to control the mode of the DPP when $\alpha^\star$ is perturbed, as it happens during an approximate optimization of $\alpha$. However, for DPPs the mode can be expressed as a Poisson binomial random variable [H+56] based on the spectrum of $\mathbf{L}$, which can be controlled [D+64] using the approximate spectrum of $\widehat{\mathbf{L}}$. While our analysis shows that in the worst case a much more accurate $\widehat{\mathbf{L}}$ is necessary compared to Theorem 3, in practice the same $m = \Omega(k \log \frac{n}{\delta})$ seems to suffice.

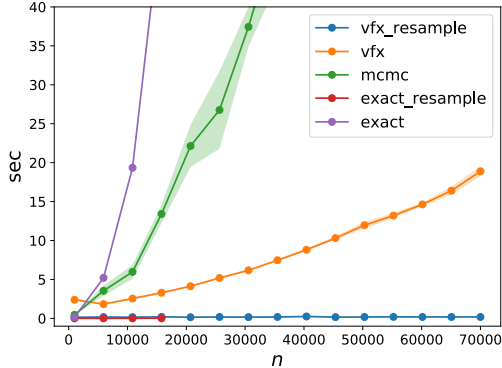
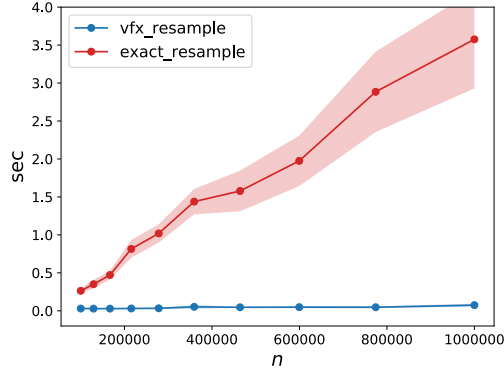

Figure 1: *First-sample* cost for DPP-VFX against other DPP samplers ($n$ is data size).

Figure 2: *Resampling* cost for DPP-VFX compared to the exact sampler of [HKP$^+$06].

## 5   Experiments

In this section, we experimentally evaluate[3] the performance of DPP-VFX and compare it to exact sampling [HKP$^+$06] and MCMC-based approaches [AGR16]. In particular, we are only interested in evaluating computational performance, i.e., how DPP-VFX and baselines scale with the size $n$ of the matrix **L**. Note that an inexact DPP sampler, e.g., MCMC samplers or previous approximate rejection samplers [Der19], also had another metric to validate, i.e., they had to empirically show that their samples were close enough to a DPP distribution. However, Section 2 proves that DPP-VFX's output is distributed *exactly* according to the DPP and therefore it is strictly equivalent to any other exact DPP sampler, e.g., the Gram-Schmidt [Gil14] or dual [KT12] samplers implemented in DPPy.

We use the infinite MNIST digits dataset (i.e., MNIST8M [LCB07]) as input data, where $n$ varies up to $10^6$ and $d = 784$. All algorithms, including DPP-VFX, are implemented in `python` as part of the DPPy library [GBV19]. All experiments are carried out on a 24-core CPU and fully take advantage of potential parallelization. For each experiment we report the mean and a 95% confidence interval over 10 runs of each algorithm's runtime.

This first experiment compares the time required to generate a first sample. We consider subsets of MNIST8M that go from $n = 10^3$ to $n = 7 \cdot 10^4$, i.e., the original MNIST dataset, and use an RBF kernel with $\sigma = \sqrt{3d}$ to construct **L**. For the Nyström approximation we set $m = 10d_{\text{eff}}(1) \approx 10k$. While this is much lower than the $\mathcal{O}(k^3)$ value suggested by the theory, as we will see it is already accurate enough to result in drastic runtime improvements over exact and MCMC. Following the strategy of Section 4, for each algorithm we control[4] the size of the output set by rescaling the input matrix **L** by a constant $\alpha^\star$ such that $\mathbb{E}[|S_{\alpha^\star}|] = 10$. Results are reported in Figure 1.

Exact sampling is performed using eigendecomposition and DPPy's default Gram-Schmidt [Gil14] sampler. It is clearly cubic in $n$, and we could not push it beyond $n = 1.5 \cdot 10^4$. For MCMC, we enforce mixing by runnning the chain for $nk$ steps, the minimum recommended by [AGR16]. However, for $n = 7 \cdot 10^5$ the MCMC runtime is 160s and exceeds the plot's limit, while DPP-VFX completes in 16s, an order of magnitude faster. Moreover, DPP-VFX rarely rejects more than 10 times, and the mode of the rejections up to $n = 7 \cdot 10^5$ is 1, i.e., we mostly accept at the first iteration.

Since in Figure 1, the resampling time of both exact and DPP-VFX are negligible, we investigate the cost of a second sample, i.e., of resampling, on a much larger subset of MNIST8M up to $n = 10^6$, normalized to have maximum row norm equal to 1. In this case it is not possible to perform an eigendecomposition of **L**, so we replace RBFs with a linear kernel, for which the input **X** represents a factorization $\mathbf{L} = \mathbf{X}\mathbf{X}^\top$ and we use an exact dual sampler on **X** [KT12]. However, as Figure 2 shows, even with a special algorithm for this simpler setting the resampling process still scales with $n$. On the other hand, DPP-VFX's complexity (after preprocessing) is light years better as it scales only with $k$ and remains constant regardless of $n$.

**Acknowledgements**

MD thanks the NSF for funding via the NSF TRIPODS program. This material is based upon work supported by the Center for Brains, Minds and Machines (CBMM), funded by NSF STC award CCF-1231216, and the Italian Institute of Technology. We gratefully acknowledge the support of NVIDIA Corporation for the donation of the Titan Xp GPUs and the Tesla k40 GPU used for this research.

## Footnotes

[2] To avoid complicating the exposition with edge cases, we assume $k \geq 1$. Note that this can be always satisfied without distorting the distribution by rescaling $\mathbf{L}$ by a constant, and is without loss of generality, as our analysis can be trivially extended to the case $0 \leq k < 1$ with some additional notation.

[3]The code used for these experiments is available at `https://github.com/LCSL/dpp-vfx`.

[4]For simplicity we do not perform the full $k$-DPP rejection step, but only adjust the expected size of the set.

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
