[Supplementary Material]

# A   Omitted proofs for the main algorithm's guarantees

In this section we present the proofs omitted from Sections 2 and 3, which regarded the correctness and efficiency of DPP-VFX. We start by showing that multiple samples drawn using the same Nyström approximation are independent.

**Lemma 5 (restated Lemma 1)** *Let $C \subseteq [n]$ be a random set variable with any distribution. Suppose that $S_1$ and $S_2$ are returned by two executions of* DPP-VFX, *both using inputs constructed from the same* $\mathbf{L}$ *and* $\widehat{\mathbf{L}} = \mathbf{L}_{\mathcal{I},C}\mathbf{L}_C^+\mathbf{L}_{C,\mathcal{I}}$. *Then $S_1$ and $S_2$ are (unconditionally) independent.*

**Proof** Let $A$ and $B$ be two subsets of $[n]$ representing elementary events for $S_1$ and $C$, respectively. Theorem 2 implies that

$$\Pr(S_1 = A \mid C = B) = \frac{\det(\mathbf{L}_A)}{\det(\mathbf{I} + \mathbf{L})} = \Pr(S_1 = A).$$

Now, for any $A_1, A_2 \subseteq [n]$ representing elementary events for $S_1$ and $S_2$ we have that

$$\Pr(S_1 = A_1 \wedge S_2 = A_2) = \sum_{B \in [n]} \Pr(S_1 = A_1 \wedge S_2 = A_2 \mid C = B) \, \Pr(C = B)$$

$$= \sum_{B \in [n]} \Pr(S_1 = A_1 \mid C = B) \Pr(S_2 = A_2 \mid C = B) \, \Pr(C = B)$$

$$= \Pr(S_1 = A_1) \Pr(S_2 = A_2) \sum_{B \in [n]} \Pr(C = B).$$

Since $\sum_{B \in [n]} \Pr(C = B) = 1$, we get that $S_1$ and $S_2$ are independent. ∎

We now bound the precompute cost, starting with the construction of the Nyström approximation $\widehat{\mathbf{L}}$.

**Lemma 6 (restated Lemma 2)** *Let $\widehat{\mathbf{L}}$ be constructed by sampling $m = \mathcal{O}(k^3 \log \frac{n}{\delta})$ columns proportionally to their RLS. Then, with probability $1 - \delta$, $\widehat{\mathbf{L}}$ satisfies the assumption of Theorem 3.*

**Proof** Let $\mathbf{L} = \mathbf{B}\mathbf{B}^\top$ and $\widehat{\mathbf{L}} = \mathbf{B}\mathbf{P}\mathbf{B}^\top$ where $\mathbf{P}$ is a projection matrix. Using algebraic manipulations we get

$$s = \mathrm{tr}(\mathbf{L} - \widehat{\mathbf{L}} + \widehat{\mathbf{L}}(\mathbf{I} + \widehat{\mathbf{L}})^{-1}) = \mathrm{tr}(\mathbf{B}(\mathbf{P}\mathbf{B}^\top\mathbf{B}\mathbf{P} + \mathbf{I})^{-1}\mathbf{B}).$$

The $\mathbf{P}\mathbf{B}^\top\mathbf{B}\mathbf{P}$ matrix in the above expression been recently analyzed by [CCL$^+$19] in the context of RLS sampling who gave the following result that we use in the proof.

**Proposition 3 (CCL$^+$19, Lemma 6)** *Let the projection matrix $\mathbf{P}$ be constructed by sampling $\mathcal{O}(k \log(\frac{n}{\delta})/\varepsilon^2)$ columns proportionally to their RLS. Then,*

$$(1 - \varepsilon)(\mathbf{B}^\top\mathbf{B} + \mathbf{I}) \preceq \mathbf{P}\mathbf{B}^\top\mathbf{B}\mathbf{P} + \mathbf{I} \preceq (1 + \varepsilon)(\mathbf{B}^\top\mathbf{B} + \mathbf{I}).$$

We decompose the condition into two parts. In the first part, we bound $k - s \le 1/2$, and then bound $z - \mathrm{tr}(\mathbf{B}\mathbf{P}(\mathbf{I} + \mathbf{B}^\top\mathbf{B})^{-1}\mathbf{P}\mathbf{B}^\top) \le 1/2$, see the proof of Theorem 3 for more details. It is easy to see that applying the construction from Proposition 3 leads to the following bound on $s$,

$$s = \mathrm{tr}(\mathbf{B}(\mathbf{P}\mathbf{B}^\top\mathbf{B}\mathbf{P} + \mathbf{I})^{-1}\mathbf{B}) \le \frac{1}{1 - \varepsilon}\mathrm{tr}(\mathbf{B}(\mathbf{B}^\top\mathbf{B} + \mathbf{I})^{-1}\mathbf{B}) = \frac{1}{1 - \varepsilon}k = k + \frac{\varepsilon}{1 - \varepsilon}k.$$

Tuning $\varepsilon = 1/(2k + 2)$, we obtain $s \le k + 1/2$ and reordering gives us the desired accuracy result. Similarly, we can invert the bound of Proposition 3 to obtain

$$(\mathbf{B}^\top\mathbf{B} + \mathbf{I})^{-1} \preceq (1 + \varepsilon)(\mathbf{P}\mathbf{B}^\top\mathbf{B}\mathbf{P} + \mathbf{I})^{-1}$$

and therefore,

$$\mathrm{tr}(\mathbf{B}\mathbf{P}(\mathbf{I} + \mathbf{B}^\top\mathbf{B})^{-1}\mathbf{P}\mathbf{B}^\top) \le (1 + \varepsilon)\mathrm{tr}(\mathbf{B}\mathbf{P}(\mathbf{I} + \mathbf{P}\mathbf{B}^\top\mathbf{B}\mathbf{P})^{-1}\mathbf{P}\mathbf{B}^\top) = (1 + \varepsilon)z \le (1 + \varepsilon)k,$$

where the last inequality is due to the fact that $\widehat{\mathbf{L}} \preceq \mathbf{L}$ since it is a Nyström approximation and that the operator $\mathrm{tr}(\mathbf{L}(\mathbf{L} + \mathbf{I})^{-1})$ is monotone. With the same $\varepsilon$ as before, we obtain the bound. Summing the two $1/2$ bounds gives us the result. ∎

Finally, we show how to compute the remaining quantities needed for DPP-VFX (Algorithm 1).

**Lemma 7 (restated Lemma 3)** *Given* $\mathbf{L}$ *and an arbitrary Nyström approximation* $\widehat{\mathbf{L}}$ *of rank* $m$, *computing* $l_i$, $s$, $z$, *and* $\widetilde{\mathbf{L}}$ *requires* $\mathcal{O}(nm^2 + m^3)$ *time.*

**Proof** Given the Nyström set $C$, let us define the matrix $\overline{\mathbf{B}} \triangleq \mathbf{L}_{\mathcal{I},C} \mathbf{L}_C^{+/2} \in \mathbb{R}^{n \times m}$ such that $\widehat{\mathbf{L}} = \overline{\mathbf{B}}\overline{\mathbf{B}}^\top$. We also introduce $\widehat{\mathbf{L}}_m \triangleq \overline{\mathbf{B}}^\top \overline{\mathbf{B}}$ to act as a $\mathbb{R}^{m \times m}$ counterpart to $\widehat{\mathbf{L}}$. Denote with $\mathbf{e}_i$ the $i$-th indicator vector. Then, exploiting the fact that $\overline{\mathbf{B}}\overline{\mathbf{B}}^\top (\mathbf{I} + \overline{\mathbf{B}}\overline{\mathbf{B}}^\top)^{-1} = \overline{\mathbf{B}}(\mathbf{I} + \overline{\mathbf{B}}^\top \overline{\mathbf{B}})^{-1}\overline{\mathbf{B}}^\top$ for any matrix, we can compute $l_i$ as

$$l_i = [\mathbf{L} - \widehat{\mathbf{L}} + \overline{\mathbf{B}}\overline{\mathbf{B}}^\top (\mathbf{I} + \overline{\mathbf{B}}\overline{\mathbf{B}}^\top)^{-1}]_{ii} = [\mathbf{L} - \widehat{\mathbf{L}}]_{ii} + \|(\mathbf{I} + \widehat{\mathbf{L}}_m)^{-1/2}\overline{\mathbf{B}}^\top \mathbf{e}_i\|_2^2.$$

Computationally, this means that we first need to compute $\overline{\mathbf{B}}$, which takes $\mathcal{O}(m^3)$ time to compute $\mathbf{L}_{C,C}^{+/2}$, and $\mathcal{O}(nm^2)$ time for the matrix multiplication. Then, $[\widehat{\mathbf{L}}]_{ii}$ is the $\ell_2$ norm of the $i$-th row of $\overline{\mathbf{B}}$ which can be computed in $nm$ time. Similarly, $\|(\mathbf{I} + \widehat{\mathbf{L}}_m)^{-1/2}\overline{\mathbf{B}}^\top \mathbf{e}_i\|_2^2$ requires $\mathcal{O}(m^3 + nm^2)$ time. To compute $s$, we simply sum $l_i$, while to compute $z$ we first compute the eigenvalues of $\widehat{\mathbf{L}}_m$, $a_i = \lambda(\widehat{\mathbf{L}}_m)_i$, in $\mathcal{O}(m^3)$ time, and then compute $z = \sum_i a_i/(a_i + 1)$. We can also recycle the eigenvalues to precompute $\log\det(\mathbf{I} + \widehat{\mathbf{L}}) = \log\det(\mathbf{I} + \widehat{\mathbf{L}}_m) = \sum_i \log(a_i + 1)$. $\blacksquare$

# B   Omitted proofs for the reduction to k-DPPs

In this section we present the proofs omitted from Section 4. Recall that our approach is based on the following rejection sampling strategy,

$$\text{sample} \quad S_\alpha \sim \text{DPP}(\alpha\mathbf{L}), \quad \text{accept if } |S_\alpha| = k.$$

First, we show the existence of the factor $\alpha^\star$ for which the rejection sampling is efficient.

**Theorem 5 (restated Theorem 4)** *There exists constant* $C > 0$ *such that for any rank* $n$ *PSD matrix* $\mathbf{L}$ *and* $k \in [n]$, *there is* $\alpha^\star > 0$ *with the following property: if we sample* $S_{\alpha^\star} \sim \text{DPP}(\alpha^\star\mathbf{L})$, *then*

$$\Pr(|S_{\alpha^\star}| = k) \geq \frac{1}{C\sqrt{k}}. \tag{1}$$

**Proof** W.l.o.g. assume that $\mathbf{L}$ is non-zero, and remember $S_\alpha \sim \text{DPP}(\alpha\mathbf{L})$ with $k_\alpha = \mathbb{E}[|S_\alpha|]$. At a high level, the proof proceeds as follows. We first prove that the probability that the size of the subset $|S_\alpha|$ is equal to its *mode* $M_\alpha$, i.e., $\Pr(|S_\alpha| = M_\alpha)$ is large enough. Then we show that varying $\alpha$ can make $M_\alpha = k$ for any $k$, and therefore we can find an $\alpha^\star$ s.t. $\Pr(|S_{\alpha^\star}| = M_{\alpha^\star}) = \Pr(|S_{\alpha^\star}| = k)$ is large enough. In other words, rescaling $\text{DPP}(\alpha\mathbf{L})$ to make sure that its mean $k_\alpha$ is close to $k$ is sufficient to guarantee that $|S_\alpha| = k$ with high enough probability.

Our starting point is a standard Chernoff bound for $|S_\alpha|$.

**Proposition 4 (PP14)** *Given any PSD matrix* $\mathbf{L}$, *if* $S_\alpha \sim \text{DPP}(\alpha\mathbf{L})$, *then for any* $a > 0$, *we have*

$$\Pr\left(\big||S_\alpha| - \mathbb{E}[|S_\alpha|]\big| \geq a\right) \leq 5\exp\left(-\frac{a^2}{16(a + 2\mathbb{E}[|S_\alpha|])}\right).$$

Note that Proposition 4 is not sufficiently strong by itself, i.e., if we tried to bound the distance $\big||S_\alpha| - \mathbb{E}[|S_\alpha|]\big|$ to be smaller than 1 we would get vacuous bounds. However, Proposition 4 implies that there is a constant $C > 0$ independent of $\mathbf{L}$ such that $\Pr\left(\big||S_\alpha| - k_\alpha\big| \geq C\sqrt{k_\alpha} + 1\right) \leq \frac{1}{2}$ for all $\alpha > 0$. In particular, this means that the mode of $|S_\alpha|$, i.e. $M_\alpha = \text{argmax}_i \Pr(|S_\alpha| = i)$ satisfies

$$\Pr(|S_\alpha| = M_\alpha) \geq \frac{1}{2C\sqrt{k_\alpha}} \sum_{i=-\lceil C\sqrt{k_\alpha}\rceil}^{\lceil C\sqrt{k_\alpha}\rceil} \Pr(|S_\alpha| = k_\alpha + i) \geq \frac{1}{4C\sqrt{k_\alpha}}. \tag{2}$$

The distribution of $|S_\alpha|$ is given by $\Pr(|S_\alpha| = i) \propto e_i(\alpha\mathbf{L})$, where $e_i(\cdot)$ is the $i$th elementary symmetric polynomial of the eigenvalues of a matrix. Denoting $\lambda_1, \ldots, \lambda_n$ as the eigenvalues of $\mathbf{L}$,

we can express the elementary symmetric polynomials as the coefficients of the following univariate polynomial with real non-positive roots,

$$\prod_{i=1}^{n}(x + \alpha\lambda_i) = \sum_{k=0}^{n} x^k e_{n-k}(\alpha\mathbf{L}).$$

The non-negative coefficients of such a real-rooted polynomial form a unimodal sequence (Lemma 1.1 in [Bra14]), i.e., $e_0(\alpha\mathbf{L}) \leq \cdots \leq e_{M_\alpha}(\alpha\mathbf{L}) \geq \cdots \geq e_n(\alpha\mathbf{L})$, with the mode (shared between no more than two positions $k, k+1$) being close to the mean $k_\alpha$: $|M_\alpha - k_\alpha| \leq 1$ (Theorem 2.2 in [Bra14]). Moreover, it is easy to see that $M_0 = 0$ and $M_\alpha = n$ for large enough $\alpha$, so since the sequence is continuous w.r.t. $\alpha$, for every $k \in [n]$ there is an $\alpha^\star$ such that $\Pr(|S_{\alpha^\star}| = k) = \Pr(|S_{\alpha^\star}| = M_{\alpha^\star})$ (every $k$ can become one of the modes). In light of (2), this means that

$$\Pr(|S_{\alpha^\star}| = k) \geq \frac{1}{4C\sqrt{k_{\alpha^\star}}} \geq \frac{1}{4C\sqrt{k+1}},$$

where the last inequality holds because $|k - k_{\alpha^\star}| \leq 1$. ∎

Finally, we show how to find $\alpha^\star$ efficiently.

**Lemma 8** *If $k \geq 1$ there is an algorithm that finds $\alpha^\star$ in $\mathcal{O}(n \cdot \text{poly}(k))$ time.*

**Proof** In order to leverage Theorem 4, we need to find an $\alpha^\star$ such that $k = M_{\alpha^\star}$, that is such that the mode of $\text{DPP}(\alpha^\star\mathbf{L})$ is equal to $k$. Unfortunately simple unimodality is not sufficient to control $M_{\alpha^\star}$ when $\alpha^\star$ is perturbed, as it happens during an approximate optimization of $\alpha$. We now characterize more in detail the distribution of $|S_\alpha|$.

In particular, $|S_\alpha|$ can be defined as the sum of Bernoullis $|S_\alpha| = \sum_{i=1}^{n} b_{\alpha,i}$ each distributed according to $b_{\alpha,i} \sim \text{Bernoulli}(\lambda_i(\alpha\mathbf{L})/(1 + \lambda_i(\alpha\mathbf{L})))$ [HKP+06]. The sum of independent but not identically distributed Bernoullis is a so-called Poisson binomial random variable [H+56]. More importantly, the following result holds for Poisson binomial random variable.

**Proposition 5 (D+64, Theorem 4)** *Given a Poisson binomial r.v. $|S_\alpha|$ with mean $k_\alpha$, let $k \triangleq \lfloor k_\alpha \rfloor$. The mode $M_\alpha$ is*

$$M_\alpha = \begin{cases} k & \text{if} \quad k \leq k_\alpha < k + \frac{1}{k+2}, \\ k \text{ or } k+1 & \text{if} \quad k + \frac{1}{k+2} \leq k_\alpha \leq k + 1 - \frac{1}{n-k+1}, \\ k+1 & \text{if} \quad k + 1 - \frac{1}{n-k+1} < k_\alpha \leq k+1. \end{cases}$$

Therefore it is sufficient to find any constant $\alpha^\star$ that places $k_{\alpha^\star}$ in the interval $[k, k + \frac{1}{k+2})$. Unfortunately, while the formula for $k_\alpha = \sum_{i=1}^{n} \lambda_i(\mathbf{L})/(1/\alpha + \lambda_i(\mathbf{L}))$ is a unimodal function of the eigenvalues of $\mathbf{L}$ which is easy to optimize, the eigenvalues themselves are still very expensive to compute. For efficiency, we can optimize it instead on the eigenvalues of a Nyström approximation $\widehat{\mathbf{L}}$, but we have to be careful to control the error. In particular, remember that $k_\alpha = \mathbb{E}[|S_\alpha|]$ when $S \sim \text{DPP}(\alpha\mathbf{L})$, so given a Nyström approximation $\widehat{\mathbf{L}}$ we can define $s_\alpha \triangleq \text{tr}(\alpha(\mathbf{L} - \widehat{\mathbf{L}}) + \widehat{\mathbf{L}}(\widehat{\mathbf{L}} + \mathbf{I}/\alpha)^{-1})$ as a quantity analogous to $s$ from DPP-VFX. Then, we can strengthen Lemma 2 as follows.

**Lemma 9 (see also Lemma 2)** *Let $\widehat{\mathbf{L}}$ be constructed by sampling $m = \mathcal{O}((k_\alpha/\varepsilon^2)\log(n/\delta))$ columns proportionally to their RLS. Then with probability $1 - \delta$*

$$\frac{1}{1+\varepsilon}k_\alpha \leq s_\alpha \leq \frac{1}{1-\varepsilon}k_\alpha.$$

**Proof of Lemma 9** We simply apply the same reasoning of Lemma 2 on both sides. ∎

Let $(1 - \varepsilon)s_{\alpha^\star} = k$, with $\varepsilon$ that we tune shortly. Then proving the first inequality to satisfy Proposition 5 is straightforward, $k = (1 - \varepsilon)s_{\alpha^\star} \leq k_{\alpha^\star}$. To satisfy the other side we upper bound $k_{\alpha^\star} \leq (1 + \varepsilon)s_{\alpha^\star} = (1 - \varepsilon)s_{\alpha^\star} + 2\varepsilon s_{\alpha^\star}$. We must now choose $\varepsilon$ such that $2\varepsilon s_{\alpha^\star} = 1/(k + 3) < 1/(k + 2)$. Substituting, we obtain $\varepsilon = \frac{1}{2(k+3)s_{\alpha^\star}}$. Plugging this in the definition of $s_{\alpha^\star}$ we obtain that $\alpha^\star$ must be optimized to satisfy

$$s_{\alpha^\star} = \frac{2k^2 + 6k + 1}{2k + 6},$$

which we plug in the definition of $\varepsilon$ obtaining our neccessary accuracy $\varepsilon = 1/(2k^2 + 6k + 1)$. Therefore, sampling $m = \widetilde{\mathcal{O}}(k_{\alpha^\star} k^4)$ columns gives us a $s_\alpha$ sufficiently accurate to be optimized. However, we still need to bound $k_{\alpha^\star}$, which we can do as follows using Lemma 9 and $k \geq 1$,

$$k_{\alpha^\star} \leq \left(1 + \frac{1}{2k^2 + 6k + 1}\right) s_{\alpha^\star} \leq \left(1 + \frac{1}{9}\right) s_{\alpha^\star} = \frac{10}{9} s_{\alpha^\star}$$

$$\leq \frac{10}{9} \left(\frac{2k^2 + 6k + 1}{2k + 6}\right) = \frac{10}{9} \left(1 + \frac{1}{k(2k + 6)}\right) k = \frac{10}{9} \frac{9}{8} k = \frac{5}{4} k.$$

Therefore $m = \widetilde{\mathcal{O}}(k_{\alpha^\star} k^4) \leq \widetilde{\mathcal{O}}(k^5)$ suffices for the accuracy. Moreover, since $s_\alpha$ is parametrized only in terms of the eigenvalues of $\widehat{\mathbf{L}}$, which can be found in $\widetilde{\mathcal{O}}(nm^2 + m^3)$ time, we can compute an $\alpha^\star$ such that $s_{\alpha^\star} = \frac{2k^2 + 6k + 1}{2k + 6}$ in $\widetilde{\mathcal{O}}(nk^{10} + k^{15})$ time, which guarantees $k \leq k_{\alpha^\star} < k + \frac{1}{k+2}$. ∎

Finally, note that the bounds on the accuracy of $\widehat{\mathbf{L}}$ are extremely conservative. In deployment, it is much faster to try to optimize $\alpha^\star$ on a much coarser $\widehat{\mathbf{L}}$ first, e.g., for $m = \mathcal{O}(k_1)$, and only if this approach fails, then to increase the accuracy of $\widehat{\mathbf{L}}$.