[Reviews · NeurIPS 2019]

Reviewer 1



Originality. This work provides a novel algorithm for sampling from a DPP, showing empirically that it is a fast algorithm and also providing a strong technical analysis. Quality & clarity. This paper is well-organized and the content is technically solid. Significance. Algorithms for fast exact sampling from DPPs are of interest to the machine learning community to enforce diversity or to perform batch Bayesian optimization. I think many researchers will be benefited from the contributions of this manuscript.

Reviewer 2



Review of "Exact sampling of determinantal point processes with sublinear time preprocessing." Summary: the paper proposes a new algorithm for exact sampling from determinantal point processes (DPP). A DPP sampling problem involves an n x n matrix L; the probability of selecting some subset S of k <= n indices is given by the determinant of a k x k matrix subset divided by the determinant of L + I. The proposed algorithm has preprocessing which is sublinear in matrix size n and polynomial in subset size k; sampling is polynomial in k and independent of n. Previous algorithms which require eigen-decomposition or MCMC are O(n^3). The main idea is to downsample the index set using a regularzed DPP and then run a DPP on this downsample. Algorithms for both fixed and random subset size k are provided. Section 2 gives details of the proposed algorithm, and Section 3 analyzes conditions for fast sampling for random subset size k. Section 4 gives a modified algorithm for fixed subset size k, and proves that it is efficient to simply reject samples which are not size k. Section 5 provides empirical evidence that the proposed algorithm is faster than MCMC and exact baselines, for subsets of MNIST. Strengths: Theoretical results prove the speed and accuracy of the proposed sampling algorithm. Some empirical results proving that the proposed algorithm is empirically faster than previous baselines. Weaknesses: It is claimed that DPP is a "tool in machine learning and recommender systems for inducing diversity in subset selection and as a variance reduction approach" but it is not clear that this paper is relevant for our field. "Determinantal Point Processes for Machine Learning" by Alex Kulesza and Ben Taskar is cited but experimental results measuring empirical accuracy on several data sets should be provided. Can you do computational cross-validation experiments to measure the test error of your algo versus other baselines? That would make your contribution much more convincing. Also is there some metric (other than test error) for measuring the empirical accuracy of your sampling algo, relative to baselines? It would be much more convincing to see some empirical demonstration of the empirical accuracy of your "exact" algorithm, relative to previous exact and inexact algos. Comments/Suggestions: Although the text is well-written in general, the proofs are difficult to follow, because they have multiple lines of math without corresponding comments. It would help to have a line of commentary/explanation for each line of math in your proof. Line 203: Le -> Let. Table 1 is a very informative summary of the time complexity of previously proposed algorithms. The main result (Theorem 1) in Section 1 is somewhat unusual, and redundant with the statements in the text/abstract. Perhaps delete or move later in the paper? It is good that error bands over 10 runs are shown in Figure 1. However linear axes are used, which makes it difficult to see differences between algorithms -- it is not clear whether or not the bottom line is linear or constant. Log-log axes would clarify. Also the orange/blue labels should be changed in either the left or the right plot, because the legend entries are inconsistent, but could be consistent (e.g. change vfx_resample to blue on the right, and change exact_resample to red). On lines 54--55, what is \bar S and \tilde S? They are not defined.

Reviewer 3



This paper studies the problem studying complexity of sampling from of a detrimental point process and proposes efficient algorithms significantly improving state-of-the-art. The paper is very well written. The comment I have is regarding the readability of the paper: may be discussing a use-case and some motivation early in the introduction will make this paper more accessible to a non-expert audience. It may also be useful if the authors discuss other more efficient repulsive point processes based on Poisson Disk Sampling (PDS) methods: 1)Zhang, Cheng, et al. "Active mini-batch sampling using repulsive point processes." arXiv preprint arXiv:1804.02772 (2018). 2)Kailkhura, Bhavya, et al. "A spectral approach for the design of experiments: Design, analysis and algorithms." The Journal of Machine Learning Research 19.1 (2018): 1214-1259. ************************ Response to the rebuttal**************************** I thank the authors for their rebuttal. I am happy with the authors’ response on my suggested changes—motivation and citations. However, I will also encourage authors to include at least one experimental comparison with DPP on the real data.

[Author Response · NeurIPS 2019]

We agree with R3 and R4's suggestion to expand our discussion of motivation and use-cases for determinantal point processes (DPP) as a tool in machine learning (ML). In the camera ready version we will stress and clarify that DPP sampling is already a well established tool in the field of ML.

DPP sampling has found applications in core ML problems such as stochastic optimization [20], data summarization [3], Gaussian processes [2], recommender systems [11], and many more. Its effectiveness has been verified repeatedly at top ML venues. Including only recent years (due to space), it has seen success at ICML'16 [14], ICML'17 [9, 12, 19], ICML'18 [3] and ICML'19 [2, 7, 10, 18], UAI'17 [20], AAAI'17 [8] and AAAI'19 [21], NeurIPS'16 [13, 15, 17], NeurIPS'17 [5, 16], NeurIPS'18 [1, 4, 6, 11], and even more successful applications of DPP sampling are likely to be present in NeurIPS'19.

We will also clarify how our approach directly impacts all of these applications. Brief summary: First, we rigorously *proved* that the samples returned by DPP-VFX are distributed *exactly* as defined by the DPP distribution. Therefore, from a black-box point of view (i.e., the point of view of the applications mentioned), samples drawn from a DPP or returned from DPP-VFX are indistinguishable, and retain the effectiveness highlighted in the literature [2, 3, 11, 20].

As demonstrated in the paper both theoretically and empirically, our method of implementing this black box provides a *significant* speed-up over existing approaches (often by many orders of magnitude), to the point where it makes large-scale application of DPPs feasible when previously it was not.

We will also address the concerns shared by R1 and R3 regarding the colors of the graphs (thanks for catching that!).

We also thank R4 for the references on Poisson Disk sampling, we will include them in the discussion of existing works.

**Addressing specific comments from Reviewer 3**

> *"It is claimed that [...] but it is not clear that this paper is relevant for our field."*

We strongly disagree with this remark, because

1. the relevance of black-box DPP samplers to ML is established by a large body of research (see above), an ICML'19 workshop and a NeurIPS'18 tutorial, and
2. we provide an *exact* DPP sampler that is orders of magnitude faster than the state-of-the-art.

> *"Can you do computational cross-validation experiments to measure the test error of your algo versus other baselines?"*

The empirical accuracy and effectiveness of DPP sampling is well established for many ML tasks (again, see the references above). Our algorithm, DPP-VFX, is simply a *very fast and exact implementation* of a DPP sampler. Therefore it would output the same samples as the other exact DPP sampler baselines but faster.

> *"Also is there some metric (other than test error) for measuring the empirical accuracy of your sampling algo, relative to baselines?"*

The most appropriate metrics to evaluate for DPP-VFX is sampling speed, and we rigorously validated it empirically, showing that it significantly outperforms state-of-the-art baselines. Note that previous *approximate* sampling methods also had another metric to validate, i.e. they had to empirically show that their samples were close enough to a DPP distribution. However we *prove* that DPP-VFX's samples are distributed *exactly* according to the DPP, and do not need to measure this metric empirically, since any negative empirical results would just be due to experimental error.

## References

[1] V.-E. Brunel. Learning signed determinantal point processes through the principal minor assignment problem. In *NeurIPS*. 2018.
[2] D. Burt, C. E. Rasmussen, and M. Van Der Wilk. Rates of convergence for sparse variational Gaussian process regression. In *ICML*, 2019.
[3] E. Celis, V. Keswani, D. Straszak, A. Deshpande, T. Kathuria, and N. Vishnoi. Fair and diverse DPP-based data summarization. In *ICML*, 2018.
[4] L. Chen, G. Zhang, and E. Zhou. Fast greedy map inference for determinantal point process to improve recommendation diversity. In *NeurIPS*. 2018.
[5] M. Dereziński and M. K. Warmuth. Unbiased estimates for linear regression via volume sampling. In *NeurIPS*, 2017.
[6] M. Dereziński, M. K. Warmuth, and D. Hsu. Leveraged volume sampling for linear regression. In *NeurIPS*. 2018.
[7] M. Elfeki, C. Couprie, M. Riviere, and M. Elhoseiny. GDPP: Learning diverse generations using determinantal point processes. In *ICML*, 2019.
[8] M. Gartrell, U. Paquet, and N. Koenigstein. Low-rank factorization of determinantal point processes. In *AAAI*, 2017.
[9] G. Gautier, R. Bardenet, and M. Valko. Zonotope hit-and-run for efficient sampling from projection DPPs. In *ICML*, 2017.
[10] J. Gillenwater, A. Kulesza, Z. Mariet, and S. Vassilvtiskii. A tree-based method for fast repeated sampling of determinantal point processes. In *ICML*, 2019.
[11] J. A. Gillenwater, A. Kulesza, S. Vassilvitskii, and Z. E. Mariet. Maximizing induced cardinality under a determinantal point process. In *NeurIPS*. 2018.
[12] I. Han, P. Kambadur, K. Park, and J. Shin. Faster greedy MAP inference for determinantal point processes. In *ICML*, 2017.
[13] T. Kathuria, A. Deshpande, and P. Kohli. Batched gaussian process bandit optimization via determinantal point processes. In *NeurIPS*. 2016.
[14] C. Li, S. Jegelka, and S. Sra. Fast dpp sampling for nystrom with application to kernel methods. In *ICML*, 2016.
[15] C. Li, S. Sra, and S. Jegelka. Fast mixing markov chains for strongly rayleigh measures, dpps, and constrained sampling. In *NeurIPS*. 2016.
[16] C. Li, S. Jegelka, and S. Sra. Polynomial time algorithms for dual volume sampling. In *NeurIPS*, 2017.
[17] Z. E. Mariet and S. Sra. Kronecker determinantal point processes. In *NeurIPS*. 2016.
[18] A. Rezaei and S. O. Gharan. A polynomial time MCMC method for sampling from continuous determinantal point processes. In *ICML*, 2019.
[19] J. Urschel, V.-E. Brunel, A. Moitra, and P. Rigollet. Learning determinantal point processes with moments and cycles. In *ICML*, 2017.
[20] C. Zhang, H. Kjellström, and S. Mandt. Determinantal point processes for mini-batch diversification. In *UAI*, 2017.
[21] C. Zhang, C. Öztireli, S. Mandt, and G. Salvi. Active mini-batch sampling using repulsive point processes. In *AAAI*, 2019.


[Meta-Review · NeurIPS 2019]

The theoretical contributions of the paper is solid and reviewers agreed that it deserves to be published on its own. The reviewed however were disappointed that a rigorous empirical validation of the theoretical claims should have been easy to demonstrate and would have made the paper stand out. Authors are encouraged to provide such validation in the final version of the paper.